# A Reassessment of the Therapeutic Potential of a Dopamine Receptor 2 Agonist (D2-AG) in Endometriosis by Comparison against a Standardized Antiangiogenic Treatment

**DOI:** 10.3390/biomedicines9030269

**Published:** 2021-03-08

**Authors:** Miguel Á. Tejada, Ana I. Santos-Llamas, María José Fernández-Ramírez, Juan J. Tarín, Antonio Cano, Raúl Gómez

**Affiliations:** 1Research Unit on Women’s Health-Institute of Health Research, INCLIVA, 46010 Valencia, Spain; anais_santos91@yahoo.es (A.I.S.-L.); juan.j.tarin@uv.es (J.J.T.); 2Department of Obstetrics and Gynecology, Hospital Clínico Universitario, 46010 Valencia, Spain; mafernanrami@hotmail.com; 3Department of Pediatrics and Obstetrics and Gynecology, University of Valencia, 46010 Valencia, Spain; 4Department of Cellular Biology, Functional Biology, and Physical Anthropology, University of Valencia, 46100 Burjassot, Spain; 5Department of Pathology, University of Valencia, 46010 Valencia, Spain

**Keywords:** endometriosis, heterologous mouse model, antiangiogenic, dopamine agonist, anti-VEGF

## Abstract

Dopamine receptor 2 agonists (D2-ags) have been shown to reduce the size of tumors by targeting aberrant angiogenesis in pathological tissue. Because of this, the use of a D2-ag was inferred for endometriosis treatment. When assayed in mouse models however, D2-ags have been shown to cause a shift of the immature vessels towards a more mature phenotype but not a significant reduction in the amount of vascularization and size of lesions. These has raised concerns on whether the antiangiogenic effects of these compounds confer a therapeutic value for endometriosis. In the belief that antiangiogenic effects of D2-ags in endometriosis were masked due to non-optimal timing of pharmacological interventions, herein we aimed to reassess the antiangiogenic therapeutic potential of D2-ags in vivo by administering compounds at a timeframe in which vessels in the lesions are expected to be more sensitive to antiangiogenic stimuli. To prove our point, immunodeficient (*NU/NU*) mice were given a D2-ag (cabergoline), anti-VEGF (CBO-P11) or vehicle (saline) compounds (*n* = 8 per group) starting 5 days after implantation of a fluorescently labeled human lesion. The effects on the size of the implants was estimated by monitoring the extent of fluorescence emitted by the lesion during the three-week treatment period. Subsequently mice were sacrificed and lesions excised and fixed for quantitative immunohistochemical/immunofluorescent analysis of angiogenic parameters. Lesion size, vascular density and innervation were comparable in D2-ag and anti-VEGF groups and significantly decreased when compared to control. These data suggest that D2-ags are as powerful as standard antiangiogenic compounds in interfering with angiogenesis and lesion size. Our preliminary study opens the way to further exploration of the mechanisms beneath the antiangiogenic effects of D2-ags for endometriosis treatment in humans.

## 1. Introduction

Pharmacologic treatment of endometriosis currently relies on minimizing exposure of lesions to the fueling estradiol [1]. Such approaches are incompatible with fertility and create a pseudomenopausal state, leading to side effects (i.e., bone mineral density loss, suffocation, and palpitations) associated with induction of a hypoestrogenic milieu [2]. This has raised the question of non-hormonal alternatives such as antiangiogenic compounds for therapeutic purposes [3].

Similar to tumors, lesions require formation of neovessels to support the continuous growth of pathologic tissue. Under this rationale, the use of commercial oncological drugs neutralizing the signaling pathway of VEGF/VEGFR2, the main driver of the angiogenic orchestra [4,5], has been explored in preclinical models of endometriosis [6]. An overall decrease in vascularization, sometimes associated with a decrease in lesion size, was reported in most studies. However, translatability of commercial VEGF/VEGFR2 blockers to the clinic for the treatment of endometriosis has been vetoed due to the unacceptable toxicity of oncological drugs for these types of patients [7]. Moreover, “physiological” angiogenesis is required for relevant biological processes, such as wound healing and fertility, which are impaired upon blockage of VEGF/VEGFR2 signaling [8,9,10,11,12,13,14]. Therefore, transferability to the clinic requires devising selective endometriotic endothelium targeting therapies, while minimizing potential harm to physiologically normal endothelium.

Dopamine receptor 2 agonists (D2-ag) are drugs with a benign clinical profile indicated for hyperprolactinemia, and do not seem to interfere with physiological angiogenesis in reproductive organs [15,16,17]. When administered in mouse cancer models these compounds reduce tumor size through partial inhibition of VEGF/VEGFR2-mediated angiogenesis in malignant tissue [18]. In the belief that D2-ags might exert selective inhibitory actions on “pathological/aberrant” angiogenesis taking place in lesions, our group explored their therapeutic potential in a well-established heterologous mouse model of endometriosis [19,20]. In such a model, pharmacological interventions started three weeks after pieces of human endometrium had been ectopically implanted in the peritoneal cavity of immunodeficient mice. After 22 days of treatment, D2-ag exerted a significant decrease in the number of immature vessels but also a significant increase in the number of mature vessels, with the net effects on overall vascularization being non-significant. Pioneer [21] and subsequent studies from Novella et al. [22,23] showed a similar pattern. Authors reported that D2-ags promoted maturation of vessels but a non-significant reduction in overall vascularization and lesion size. These results differ from the dramatic decreases in vascularization and lesion size observed in similar experiences in which specific VEGF/VEGFR2 blocker compounds were used [6]. A pilot study in humans [24] added further confusion to this issue as the D2-ag quinagolide was shown to decrease the size of peritoneal lesions but the amount of vascularization appeared unaffected. Such a scenario raises concerns as to whether the exploration of the antiangiogenic pathways transduced by D2-ag deserve further attention for therapeutic purposes in the context of endometriosis. We reasoned that confirming the capacity of D2-ags to exert antiangiogenic and associated effects on lesion size at a significant level in endometriosis might be the first step to warrant further exploration of such issues.

However, looking back over our experience, we realize that the experimental design in ours [19] and previous seminal works [21,22,23] might have not been optimal to detect actual antiangiogenic events. In this regard, the timing of D2-ag administration was not aligned with the window of time in which endometrial vessels in the xenograft of the mouse model are more sensitive to pro/antiangiogenic stimuli [25]. The experimental design did not include a standardized VEGF/VEGFR2 treatment group so that the actual significance of antiangiogenic effects exerted by D2-ags might be compared. Finally, the methodology employed to estimate the volume of lesions (i.e., use of calipers) was not sensitive enough to detect variations at the appropriate scale. For the above reasons, it was plausible that the effects of D2-ags (if any) on angiogenesis and lesion size might have been neglected/masked in our seminal work. Therefore, we sought to reassess the potential therapeutic effects of D2-ags in the heterologous mouse model by setting an experimental design in which the above mentioned “pitfalls” had been fixed. For such purposes, (a) the timing of pharmacological interventions was set up to overlap with the period in which the sensitivity of lesions to angiogenic stimuli is enhanced; (b) lesion size was assessed by real-time quantification of fluorescence emitted by previously labelled xenografted tissue; (c) evaluation of the effects of D2-ags on the heterologous mouse model was compared against CBO-P11, a standardized VEGF blocker [26]. Additionally, we took advantage of this effort to further explore the effects of D2-ag on neuroangiogenesis in the mouse model by assessing nerve fiber density. Overall results obtained are shown and discussed as follows.

## 2. Materials and Methods

### 2.1. Experimental Animals

The experiments were carried out in a total of 24 athymic, 5 week-old, nude female mice (*NU/NU Nude Mouse (Crl:NU-Foxn1^nu^*)), Charles River, Barcelona, Spain). All animals were housed in specific pathogen-free conditions at 26 °C with a 12-h light–12-h dark cycle and fed ad libitum. The study was approved by the Institutional Animal Care Committee at the University of Valencia (14/010, 2014), and all procedures were performed following the guidelines for the care and use of mammals from the National Institutes of Health.

### 2.2. Heterologous Mouse Model of Endometriosis

#### 2.2.1. Preparation of Endometriosis Recipient Mice

To avoid cycle-dependent variations, mice were ovariectomized and hormone levels were normalized by placing 60-day-release capsules containing 18 mg of 17β-E2 (Innovative Research of America, Sarasota, FL, USA) under the neck skin. Mice were left to rest for 11–12 days after surgery without further intervention until being implanted with mCherry-labeled human endometrium fragments (3–5 mm^3^ in size).

#### 2.2.2. Obtention and Preprocessing of Human Endometrial Biopsies

Use of human tissue specimens was approved by the Institutional Review Board and Ethics Committee of INCLIVA (2017-253, 2017). Human eutopic endometrial tissue at the edge of the late proliferative/early secretory phase was acquired from egg donors at the time of oocyte retrieval after written informed consent, in accordance with the Declaration of Helsinki. The endometrial biopsies were obtained following methodology previously described [27]. The human tissue was immediately harvested and put in maintenance M199 medium with 10% fetal bovine serum (FBS), 10 mmol/L N-2-hydroxyethylpiperazine-N0-2-ethanesulfonic acid (HEPES) solution and antibiotic/antimycotic solution (Gibco, Carlsbad, CA, USA). Fragments were sited on a 10-cm Petri dish, cut into 5–10 mm^3^ pieces with a scalpel, and punctured carefully with a 24-G syringe to augment the quantity of tissue surface exposed. Approximately 40–50 strip fragments were obtained from each biopsy. Next, tissue pieces were placed in 96-well plates and washed twice with antibiotic-free M199 medium.

#### 2.2.3. Labeling of Human Endometrial Preprocessed Biopsies

Endometrial tissue was labeled as previously described [27,28]. Briefly, endometrial fragments were incubated with 1 × 10^8^ PFU/mL Ad-mCherry (Vector Labs, Burlingame, CA, USA) diluted in antibiotic-free DMEM F-12 overnight (12–18 h) at 37 °C, with 5% CO_2_ inside an incubator with gentle agitation. Tissue fragments were then rinsed with DMEM F-12 twice, and its fluorescence was observed in the red channel (568 nm) with the use of an inverted microscope (Eclipse; Nikon, Tokyo, Japan). Fragments with optimal fluorescent intensity were preselected for subsequent engraftment.

#### 2.2.4. Implantation of Labeled Endometrial Fragments in Recipient Mice

Under inhalation isoflurane anesthesia, mouse peritoneum was accessed through a small incision made in the middle of the abdomen. A total of 2–3 Ad-mCherry labeled endometrial tissue strips were fixed in the peritoneum of each animal with the use of an N-butyl-ester cyanoacrylate adhesive (3M Animal Care, Minneapolis, MN, USA).

### 2.3. Pharmacologic Interventions during the Time Course

After implantation surgery, animals were randomly divided into three groups (*n* = 8 each group) with pharmacological interventions starting five days later. The D2-ag group was treated every three days with cabergoline (50 µg/kg by oral gavage, Pfizer, Valencia, Spain); the anti-VEGF group with daily intraperitoneal administration of 0.6 mg/kg CBO-P11 (Sigma, Barcelona, Spain), and the control group with 100 µl of 5% glucosaline vehicle orally. Treatments were given over 3 weeks. The concentrations of the VEGF inhibitor (CBO-P11) have been previously assayed by our group in the heterologous model of endometriosis and shown to reduce overall vascularization when administration started 1 week after implantation surgery [27]. Dose was originally chosen as that able to exert maximal reduction on vascularization free of the main side effects in a glioma tumor model [29]. The D2-ag (cabergoline) concentration mimicked the one assayed in the previous report from our group in the heterologous mouse model [19], which was shown not to reduce lesion size and vascularization when administration started 3 weeks after implantation surgery. Dose was set up as optimal on the basis of dose response experiments performed on previous works by our group [16,19].

### 2.4. Non-Invasive Assessment of Lesion Size

#### 2.4.1. Obtention of Raw Fluorescent Signal

The extent and intensity of the fluorescence emitted by the labeled fragments served to indirectly estimate the size of the lesion during the treatment period, as previously described [27,30]. Noninvasive observation of fluorescence was achieved with the IVIS Spectrum Preclinical In Vivo Imaging System (Perkin-Elmer, Madrid, Spain) and related software (Living Image 4.7.3, Perkin-Elmer, Madrid, Spain) coupled to an isoflurane gas anesthesia machine (XG-8 Gas Anesthesia System; Xenogen, Perkin-Elmer, Madrid, Spain). Monitoring of fluorescent intensity was performed every two or three days over a period of 22 days. Immunofluorescence images were attained by means of epiluminescence with peak absorption/emission pair filters set at 587 nm and 610 nm, respectively. The field of view was set at 10 cm, the minimal number of counts was set to 6000 and the exposure time was set in function of the time required to reach the minimal number of counts (usually 0.25 and 2 s at the beginning and end, respectively of the time course) as previously described [31].

#### 2.4.2. Quantification of Lesion Fluorescent Signal

Images were presented as false-color photon-counts superimposed on a grayscale anatomic image with the optical intensity (photon flux) expressed as the average radiant efficiency in photons/s/cm^2^, as previously described [27,32]. In order to detect differences throughout the treatment period, signal intensity was normalized in each animal through the time course by setting a fixed period of exposure. Regions of interest (ROIs) equivalent to lesions were automatically established by the software after setting a threshold over the lesion with minimal intensity.

### 2.5. Immunofluorescence and Immunohistochemistry

After 21 days of treatment, animals were sacrificed with CO_2_ in a sealed chamber, the peritoneal cavity was accessed, and the implants were collected. For immunohistochemical purposes, fragments were fixed in 4% PFA overnight for 48 h at 4 °C, before being embedded in paraffin wax. For immunofluorescent analysis, implants were fixed with PFA 4%, cryoprotected in 30% sucrose phosphate buffer for 48 h, and frozen in OCT embedding compound. The vascularization of the endometrial implants was detected following protocols previously described by our group [19,27], employing a rabbit polyclonal antibody (1:200, ab28364, Abcam, Cambridge, MA, USA) against CD31 (a marker of endothelial cells) to stain vessels, and a mouse monoclonal antibody (1:50, S-501, Sigma, Barcelona, Spain) labeled with Cy3 to detect α-smooth muscle actin (α-SMA) in mature vessels. Proliferating cells were detected with the anti-Ki67 antibody (MIB-1, 1:100, M7240, Dako Cytomation, Gloostrupp, Denmark) and apoptotic cells were marked using an ApopTag ISOL Dual Fluorescence Apoptosis Detection Kit (DNase types I and II; Millipore, Burlington, MA, USA) as previously described [19,33,34]. Nerve fibers were detected by using antibodies against the neuronal markers PGP 9.5 (1:200 Dako Cytomation, Gloostrupp, Denmark) and Beta III tubulin (1:100, PA5-85639, Thermo Fisher Scientific, Waltham, MA, USA) by respectively employing immunohistochemical [23] and immunofluorescent [35] protocols previously described.

#### Quantification of Immunofluorescent/Immunohistochemical Images

The two main cross-sections obtained in each implant were considered to be representative of the sample. Three random high-power fields were photographed per cross-section with the use of a “LAS V4” software image capture (Leica Mycrosystem, Heerbrugg, Switzerland) system related to a Nikon Eclipse E400 microscope. Therefore, a total of 6 images (1 implant x 2 cross sections x 3 high-power fields) per lesion were acquired for the subsequent determination of each parameter of interest. Quantitative analysis of cellular proliferation (Ki67 expression), vascularization (CD31 and α-SMA), nerve fiber density (PGP 9.5 and B-III tubulin) and apoptosis (terminal deoxynucleotide transferase-mediated dUTP nick-end labeling (TUNEL)) were performed by quantification of stained areas from the total sample area with Image Pro Plus (Media Cybernetics, Silver Spring, MD, USA) as previously described elsewhere [19,27,34]. Semiquantitative analysis of the percentage of mature and immature vessels was also assessed as previously described by our group [27].

### 2.6. Statistical Analysis

The data were analyzed with the SigmaPlot 12.0 program (Systat Software Inc., San Jose, CA, USA). Data were expressed as mean ± SD. Two-way analysis of variance (ANOVA) followed by the Holm–Sidak post hoc test was used to discern the effects of treatments on lesion size over time. One-way analysis of variance (ANOVA) followed by the Holm–Sidak post hoc test was used to discern the effects of treatments in vascularization, vessel maturity, proliferation, apoptosis and nerve fiber density. *p* < 0.05 was considered statistically significant.

## 3. Results

### 3.1. Assessment of Endometriotic Lesions by FLI

The amount of raw fluorescence signal emitted was comparable amongst lesions at the beginning of the experiment. Data dispersion was low and quite homogeneously distributed amongst groups. To discard unnoticed bias during the process of selecting lesions, the extent of fluorescence intensity emitted by each lesion during the time course was normalized against the time point at which the signal was maximal. Overall, the pattern of fluorescent signal emission was quite similar in all three groups over the time course (Figure 1D). In this regard, upon implantation, signal intensity increased and peaked after approximately one week (i.e., maximal intensity was observed on either Day 5 or Day 8 after implantation in all animals). Subsequently, the signal started to uniformly decline and progressively continued with that trend until the end of the experiment, in agreement with the episomal expression of Ad-mCherry virus [27,28,31]. Although the decreasing slopes were more prominent in the anti-VEGF and D2-ag groups, significant differences against the control group only became evident at the latter stages (i.e., day 18) of monitoring (Figure 1D). At the end of the study period, after 21 days of treatment, a 33.12 and 32.93% reduction in fluorescence intensity was observed in the D2-ag and anti-VEGF groups when respectively compared to the untreated control.

### 3.2. Vascular Density and Maturity in Endometriotic Lesions

In untreated lesions, the pattern of staining was in agreement with the existence of a majority of vessels of the immature type. Indeed, in the control group, more than half of the putative vessels stained green (CD31+), free of surrounding red labeled (α-SMA-) smooth muscle cells. In contrast, in both treated groups, the presence of immature vessels in lesions was scarce, and observations of green stained structures surrounded by red color identifying vessels of the mature (CD31+/α-SMA+) type were much more common (Figure 2A). Thus, the percentage of mature vasculature in the control group was a mere 40%, which was significantly lower (*p* < 0.001) than the 90% and 76% values observed in the anti-VEGF and D2-ag treated group (Figure 2B). When we assessed vascular density, we detected a 29.7% and a 34.3% reduction in the percentage of stained area in the D2-ag and anti-VEGF treated groups vs. control (*p* < 0.05; Figure 2C).

### 3.3. Proliferation

Proliferating cells were observed throughout all sections, although they tended to be especially abundant in epithelial glands (Figure 3A–C) in all groups. The percentages of proliferating Ki67 nuclear-stained cells in the endometriotic lesions of animals treated with D2-ag (2.14%) and anti-VEGF (2.27%) were slightly decreased vs. control (2.91%). This corresponds with moderate (22.2% and 26.7%) reductions in the index of cell proliferation in the anti-VEGF and D2-ag group (the latter reaching significant difference; *p* < 0.05; Figure 3D) when compared to control.

### 3.4. Apoptosis

Apoptotic cells were overall scarce and not especially abundant in either of the groups assessed. Quantification with software image analysis revealed a dramatic increase in this parameter in both the D2-ag (341%) and anti-VEGF treated (405%) groups when compared to control (** *p* < 0.001; Figure 4D).

### 3.5. Nerve Fiber Density

In several images (18%), the pattern of expression PGP 9.5 was in agreement with the shape of staining expected for nerve fibers identified as narrow/thin structures. However, for most of the images taken (82%), PGP 9.5 also stained against a considerable number of whole cells in the xenografted tissue (see Appendix A). Provided that the staining pattern of PGP 9.5 was not exclusive for nerve fibers, we refrained from further quantitative analysis of this marker. In contrast, the Beta III tubulin staining pattern was consistent with longitudinal and transverse sections of nerve fibers (Figure 5A–C) with scarce labeling mostly represented by spotty round and narrow longitudinal bands. The signal appeared quite specific and was observed in all three groups. Overall, Beta III tubulin staining was significantly higher in the control when compared to D2-ag and anti-VEGF treated groups (*p* < 0.001; Figure 5D).

## 4. Discussion

Targeting global angiogenesis was anticipated 50 years ago by Judah Folkman to exert therapeutic effects in tumors [36]. With such a rationale, antiangiogenic drugs, originally designed for oncological purposes, have been assayed in preclinical models of endometriosis and shown to reduce the size of lesions [37,38]. These compounds however interfere with reproductive angiogenesis [39,40] and non-specifically target physiological and quiescent vasculature, thus exerting serious side effects, such as thromboembolism and gastrointestinal perforation, which are unacceptable for endometriosis patients [7,41,42]. In order to overcome these limitations, we proposed the use of D2-ags as more benign potential selective inhibitors of pathological angiogenesis to treat endometriosis in humans [24]. However, when assayed in a heterologous mouse model of endometriosis, none of the D2-ag compounds tested were shown to significantly decrease overall vascularization or lesion size [19,21,22,23]. In spite of the claims of antiangiogenic effects, the fact is that significant effects were restricted to a reduction in the number of immature vessels, which was counterbalanced by an increase in the number of mature vessels. Turning vessels insensitive to angiogenic stimuli by inducing its maturation (i.e., surrounding them with pericytes and smooth muscle cells) might hinder angiogenesis but is not a direct proof of vessel destruction compromising tissue survival. However it is noteworthy to mention that despite the net effects on total vascularization being non-significant, the above mentioned studies agreed in reporting a tendency of D2-ag to reduce overall vascularization, thus suggesting that actual antiangiogenic effects might be taking place at a moderate extent. In line with this, although the pioneer study by Novella et al. [21] did not detect any change in lesion size, they reported a significant reduction in the number of glands in response to D2-ag, suggesting the existence of events leading to tissue destruction. To uncover the antiangiogenic effects suggested by previous studies, we reassessed the outcome of D2-ag in a refined heterologous mouse model in which the effects on lesion size might also be adequately measured. In addition, we included an antiangiogenic reference group in such a way that the extent of the effects might be standardized. To maximize the chances of detecting antiangiogenic effects, pharmacological interventions were aligned with the timing in which xenograft vessels have been described to be more sensitive to angiogenic stimuli in the heterologous mouse model [43]. In such studies, vascularization in lesions was profoundly reduced (i.e., 40–65% decrease) if antiangiogenic drugs were administered concomitantly to graft implantation but not (i.e., 2–5% decrease) if given three weeks later [44,45]. With a similar approach, we herein uncovered that D2-ag administration a few days after graft implantation is able to reduce overall vessel density and size of lesions in the mouse model. By using such an approach, we were able to detect the significant differences in overall vascularization and lesion size that previous reports [19,21,22,23] had been unable to detect. In agreement with previous studies, we also observed the transition of vessels from the immature to the mature type. The mechanism mediating the maturation of vessels through D2-ag has been well described at the molecular level [46,47].

We speculate that a plausible explanation for the more dramatic effects of D2-ag in overall vascularization observed in our study compared with previous reports might depend on the different stages of maturation of vessels at the timing of drug administration. The classical criterion for definition of immatureness is the absence of alpha-SMA (alpha-SMA) surrounding vessels. According to comparative observations in untreated animals, the number of immature vessels seems to remain similar in lesions over the time course and does not seem different between ours and previous studies [19,21,22,23]. However recent reports have shown that α-SMA expression in pericytes, the cells “protecting” endothelial cells from angiogenic stimulus, is lost during tissue fixation [48]. In this context, the number of actual immature vessels might be overestimated at later time points of graft development, thus explaining the timing dependent effects of VEGF/VEGFR2 inhibitor observed in the lesions of the het mouse model.

In this study, we detected a significant increase in the number of apoptotic cells in treated groups vs. controls. This result would be in agreement with the decreases in lesion size observed and processes of tissue regression taking place. Neither Novella et al. [21,22,23] nor us [19] evaluated apoptosis in previous related studies so it is not possible to compare/highlight the effects resulting from the different timing of D2-ag administration. In regards to proliferation, here we report a significant decrease in D2-ag animals vs. controls in a similar fashion as observed in our previous study [19]. Our results are however discrete if compared against pioneer studies [21] from Novella et al. who reported 2–3 fold lower amounts of proliferation in treated animals than we did. We do not have any explanation for this phenomenon.

It was exciting to observe that the extent of antiangiogenic effects was similar in both the anti-VEGF and Cb2 group in our study. These results suggest that Cb2 is a powerful VEGF/VEGFR2 inhibitor in decreasing vascularization and lesion size. It is unclear however if/that D2-ag affects angiogenesis by directly interacting with VEGFR2 on vessel endothelial cells or through different mechanisms of action. Internalization (i.e., inactivation) of VEGFR2 mediated by paracrine interaction with D2, as observed in tumor tissue endothelial cells [17,49], was originally proposed [19] in this regard. However, endometriotic lesion vessels do not express D2 [42], thus precluding this possibility. Interference with migration of endothelial progenitor cells (EPCs) to newly forming vessels [50,51,52] has been proposed as a more plausible explanation. Indeed, migration of EPC peaks during the first two to three weeks after graft implantation [52]. This possibility would agree with the timing dependent antiangiogenic effects observed for D2-ag in these vs. previous studies [19,21,22,23]. Additionally, inhibition of VEGF secretion by local macrophages [47] in endometriotic lesions [24] has also been proposed. By now, none of these or other alternative potential mechanisms of action has yet been explored/confirmed to mediate D2-ag effects in endometriosis.

At some point this is paradoxical because the rationale for the development of D2-ag clinical trials in endometriosis was supported by selective inhibition of angiogenesis through mechanisms of action which are not yet understood. A pilot (second look laparoscopy) experiment in humans provided puzzling results in this regard as a D2-ag induced shrinkage of lesions, but vessel density appeared unaffected [42]. Doubts have arisen as to whether D2-ag reduced lesion size through a masked antiangiogenic mechanism, due to non-antiangiogenic mechanisms or by a combination of both. In the context of our poor knowledge on the pathophysiology of endometriosis, the rationale for performing clinical trials with D2-ag [53] is mostly empiric. Fortunately, this has not dissuaded clinicians from performing pilot studies confirming the potential therapeutic effects of D2-ag [54]. We hope that if clinical trials are concluded and results are released this will encourage us and others to perform research to dissect the molecular mechanism of action of D2-ag and refine current therapies.

In this regard, data from a pilot clinical trial assessing the effects of cabergoline in endometriosis-associated pain has been recently released, pointing to an analgesic effect of D2-ag [55]. In this context, we aimed to take advantage of our study and assess in our mouse model whether Cb2 was able to decrease nerve fiber density.

Provided that human eutopic endometrium is, as claimed to be, free of PGP 9.5 positive nerve fibers [56], we were very curious to assess whether the reinervation is as dramatic as previously reported [23]. To our surprise, PGP 9.5 started to show a dramatic abundant expression in whole stromal cells. PGP 9.5 has been reported to be a poor marker of neural cells [57], but it is commonly used for specific labeling of nerve fibers in the endometrium [56]. Therefore, it is likely that promiscuous expression of PGP 9.5 in stromal cells is enhanced by the ectopic placing of endometrial tissue, a phenomenon that will require further analysis in the future. In the meantime, and provided that PGP 9.5 is not suitable, we switched to Beta-III tubulin as a more reliable marker of nerve fibers. In this case, the process of reinervation was not as dramatic and the presence of nerve fibers was scarce in all three groups. Nerve fiber density was significantly lower in treated animals vs. control.

In addition to the above, D2-ag is speculated to exert immunomodulatory actions [24,48], which might contribute to reducing lesion size and associated pain. However, due to their immunosuppressed condition, the mouse model we employed does not provide an adequate scenario to assess the effects of drugs on the inflammatory profile. Such tasks would better be implemented in an immunocompetent homologous model [58], as recently performed by Kim et al. [59].

Interpretation of results in animal models is complicated, due to their inherent limitations. With this work, we have learnt that if we aim to dissect the antiangiogenic contribution of D2-ag in endometriosis in this improved animal model, the timing of D2-ag administration is key for such purposes. In this regard, there has been a tendency to let lesions establish in the model for weeks before pharmacological interventions in the belief that such a situation better reflects the angiogenesis taking place in established lesions or the clinical situation in humans. However, there is no clear scientific rationale for such a strategy. In conclusion, herein we have shown that under adequate timing of administration, D2-ag significantly reduces overall vascularization, leading to shrinkage of lesions implanted in the heterologous mouse model. We hope these observations will boost efforts to characterize the molecular mechanism of action of D2-ag and will encourage further clinical trials to assess the effects of these compounds in endometriosis.

## Figures and Tables

**Figure 1 biomedicines-09-00269-f001:**
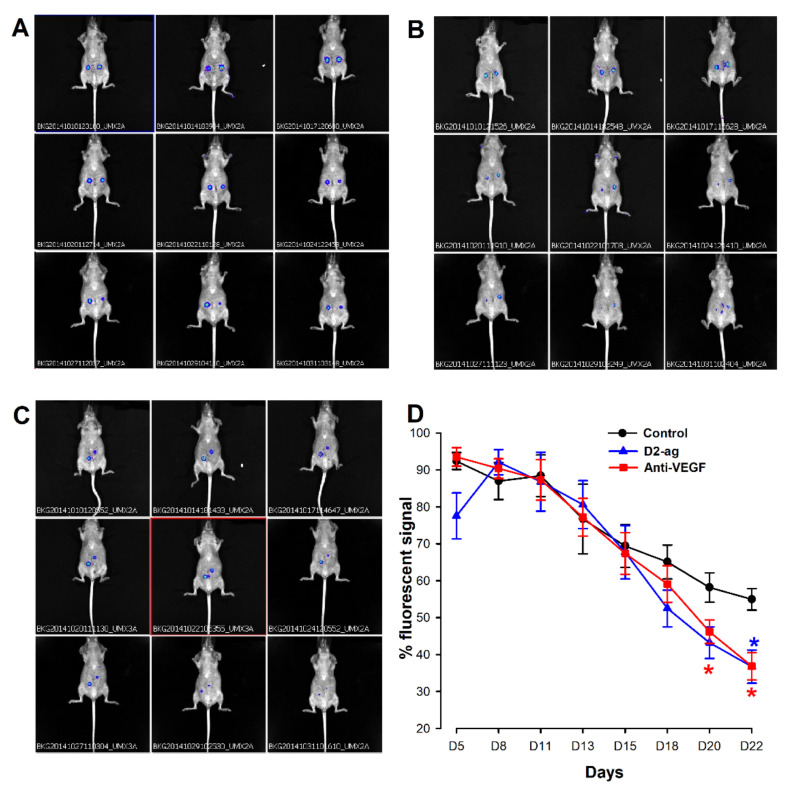
In vivo imaging of fluorescence emitted by endometriotic implants in control, D2-ag (cabergoline) and anti-VEGF (CBO-P11) treated animals. Images show illustrative examples of the fluorescence signaling provided by mCherry-labeled endometriotic implants during monitoring throughout the treatment time course in a representative animal of the control (**A**,**B**) D2-ag and (**C**) anti-VEGF groups. (**D**) Graph shows quantitative analysis of normalized average fluorescence intensity provided by implants in the three groups during the time course (mean ± SD). Raw fluorescence signal was normalized to the time point at which the signal (usually Day 5 or Day 8) was maximal. * *p* < 0.05 vs. control group in each time.

**Figure 2 biomedicines-09-00269-f002:**
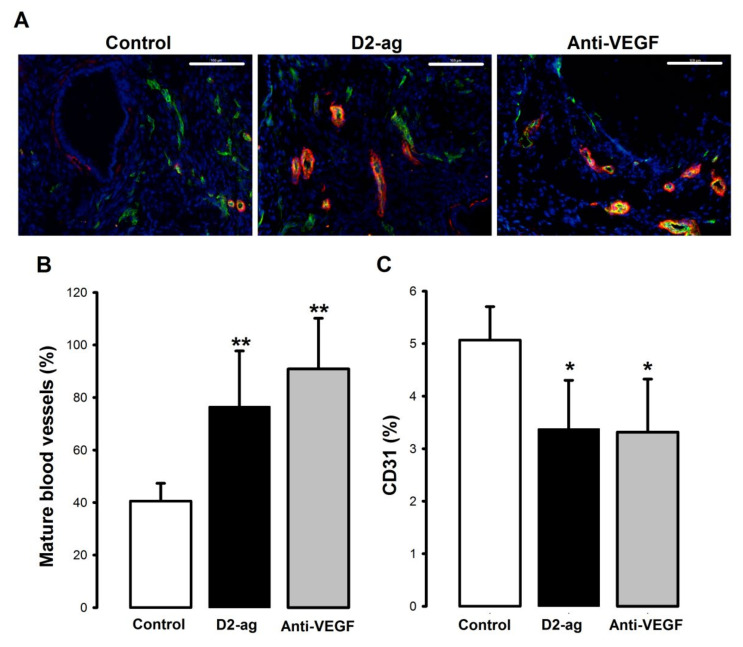
Effects of pharmacological interventions on vessel density and vascularization patterns of endometriotic implants. (**A**) Shows representative staining patterns for CD31 (endothelial marker, green color) and α-SMA (muscular layer marker, red color) in endometriotic implants of vehicle (control), D2-ag or anti-VEGF treated mice. Mature vessels surrounded by α-SMA are easily identified by overlapping yellow color. Note the increased number of yellow vessels in D2-ag and anti-VEGF treated vs. control groups. Also note the increased number of immature green vessels devoid of α-SMA in controls. Scale bar = 100 µm. (**B**) Graph shows the percentage (mean ± SD) of mature vessels as represented by the number of vessels surrounded by a red muscular layer (CD31+/α-SMA+) divided by the total number of vessels (CD31+/α-SMA-) +(CD31+/α-SMA+). ** *p* < 0.001 vs. control group. (**C**) Graph shows the vascular density of endometriotic implants as represented by the percentage (mean ± SD) of CD31 stained area vs. total area. * *p* < 0.05 vs. control group.

**Figure 3 biomedicines-09-00269-f003:**
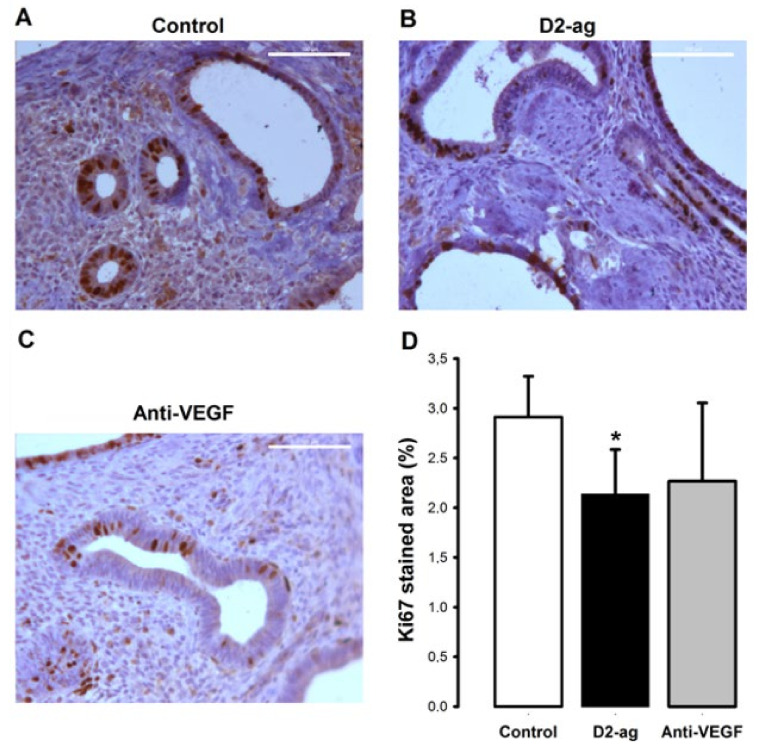
Effect of D2-ag and anti-VEGF on cell proliferation of endometriotic implants. Pictures correspond to representative images of proliferating cells stained against the proliferation marker Ki67 (brown color) in endometriotic implants of (**A**) control, (**B**) D2-ag and (**C**) anti-VEGF treated mice at sacrifice. Scale bar = 100 µm. (**D**) Graph corresponds to cellular proliferation among the three groups (mean ± SD). * *p* < 0.05 vs. control group.

**Figure 4 biomedicines-09-00269-f004:**
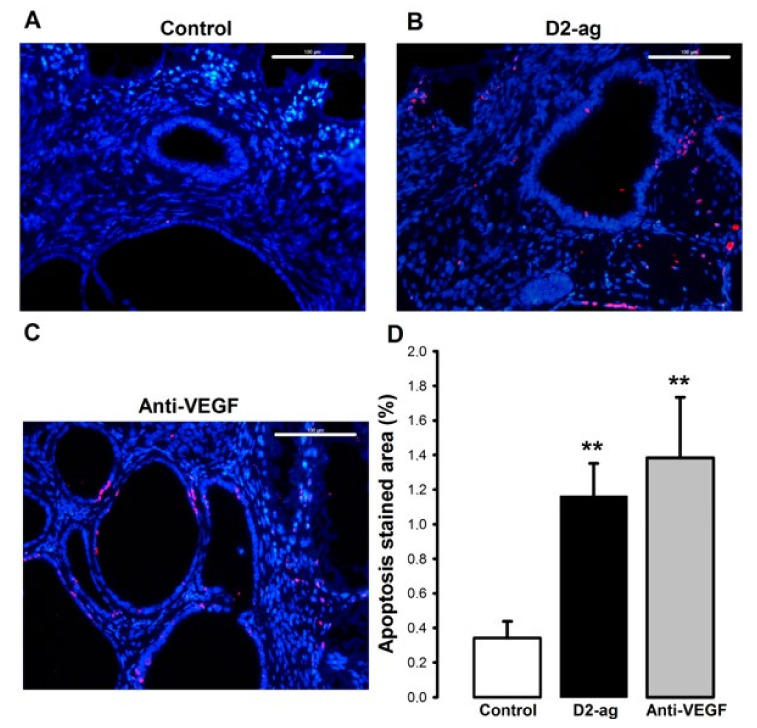
Effect of D2-ag and anti-VEGF treatments on apoptosis of endometriotic implants. Pictures correspond to representative images of apoptotic cells recognized by TUNEL (red color) in endometriotic implants of (**A**) control, (**B**) D2-ag and (**C**) anti-VEGF groups. Scale bar = 100 µm. (**D**) Graph shows apoptotic stained area normalized against DAPI area (mean ± SD). ** *p* < 0.001 vs. control group.

**Figure 5 biomedicines-09-00269-f005:**
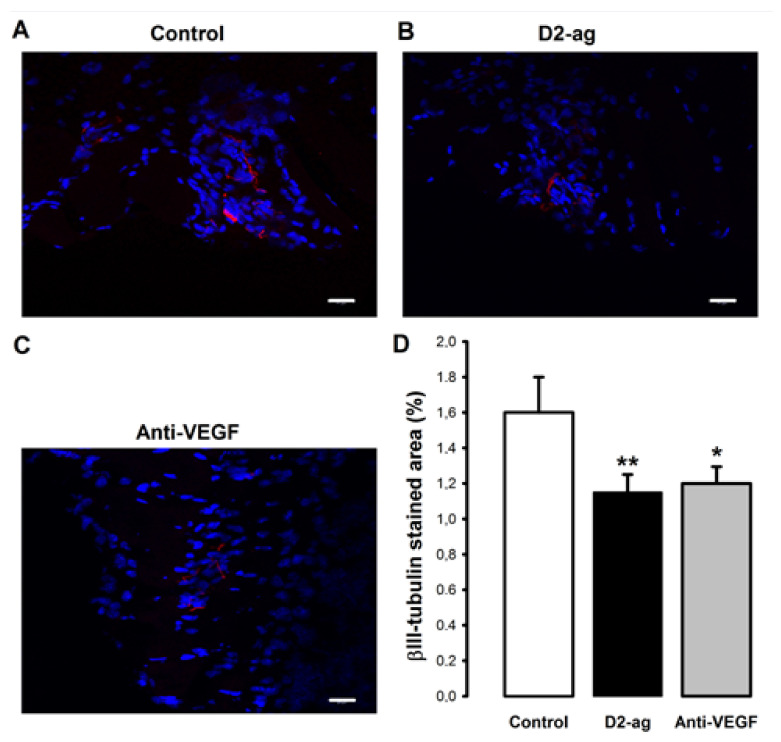
Pictures correspond to representative images of nerve fibers recognized by Beta III tubulin staining (red color) in (**A**) control, (**B**) D2-ag and (**C**) anti-VEGF groups. Scale bar = 20 µm. (**D**) Graphcorresponds to quantification of nerve fibers among the three groups (mean ± SD). **p* < 0.05, ***p* < 0.001 vs. control group.

## Data Availability

Not applicable for this study.

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
