# Peer review of "A Reassessment of the Therapeutic Potential of a Dopamine Receptor 2 Agonist (D2-AG) in Endometriosis by Comparison against a Standardized Antiangiogenic Treatment"

_biomedicines, 2021, doi:10.3390/biomedicines9030269_

Round 1
Reviewer 1 Report
As I have mentioned in my former review the paper of Santos-Llamas et al. is not fully original as it duplicates the results that has previously been reported by other investigators. The authors claim that their present study extends the above-mentioned reports showing and importance of timing in course of dopamine receptor type 2 agonist application. However, this does not add too much to the current knowledge on the possible clinical application of dopamine agonists in the treatment of endometriosis.
The authors significantly improved their manuscript, however, there are still some points which remain unanswered.
- How the numerical analysis shown in figs 2-5 was performed? How many fields were analyzed and how these fields were chosen.
- I have a concern regarding statistical analysis and data presentation as a mean and SEM. As I have previously mentioned the nature of the results (proportions) suggests that they should be rather subjected to a non-parametric statistical evaluation. Superficial analysis of SEMs shows that the standard deviation values are high that may argue for a not normal distribution of data. So, authors, please, present median and range for the analyzed groups.
There is also one more thing that requires explanation. The number of coauthors in the present version of the manuscript, which is not considerably different from the former one, is reduced. Does it mean that the authors deleted from the present version (3 names) of the manuscript did not contribute enough? If so, why they were included in the former version?
Author Response
Reviewer 1
As I have mentioned in my former review the paper of Santos-Llamas et al. is not fully original as it duplicates the results that has previously been reported by other investigators. The authors claim that their present study extends the above-mentioned reports showing and importance of timing in course of dopamine receptor type 2 agonist application. However, this does not add too much to the current knowledge on the possible clinical application of dopamine agonists in the treatment of endometriosis.
- The reviewer is partially right. The manuscript is entitled ..” A reassessment..” we aimed to refine and improve the model and experimental design to dissect and uncover actual antiangiogenic effects of D2-ag and effects on lesion size. During the process we thus reported findings which do not duplicate previous results. Therefore, we disagree with the “duplicate” statement. As detailed in our response to the editor we found significant differences (decreases) in overall vascularization and in lesion size) which had not been previously shown/clearly demonstrated (suggested or insinuated is not equal to demonstrated significantly) by previous reports (19, 21-23). In order to highlight our findings, we have clarified in our manuscript a misconception of previous reports who assumed that decreases in immature vessels do necessarily reflect antiangiogenic events. As a counterpart we better detailed results shown by previous reports in regards to overall/total vascularization, lesion size and proliferation so as to recognize merit deserved by previous reports (see response to the editor and manuscript attached) We also analyzed apoptosis (not previously assessed in previous reports). We also included a reference (anti-VEGF) group so as the extent of the antiangiogenic response of D2-ag could be estimated by comparison against a standar VEGF treatment. We also included a more reliable and more accurate methodology to estimate lesion size. We also employed a new protocol to quantify nerve fiber density in a more accurate way and thus address potential pitfalls from previous reports. All this was done with a constructive spirit recognizing the value deserved by previous reports.
We do not have an objective scale bar to measure the extent of novelty of our work and we fell reviewer 1 does not have such tool either. We feel this is more a subject related to personal views and subjective opinion and it seems ours differs from reviewer´s 1. So, we have just exposed the facts in this regard.
In regard to our contributions to endometriosis,” this does not add too much to the current knowledge…”, we do not consider that a manuscript requires to provide a clear “clinical application of dopamine agonists to the clinic” in order to be accepted for publication. In fact, we do not know of any manuscript of that kind having been published. Indeed, we would be surprised at this point provided that use of D2-ag for endometriosis is mostly empirical. We do not envision any reason why this manuscript should not be published. Perhaps the statement from reviewer 1 at this point is an unnecessary despise to our work. It is ok, as mentioned above we understand this is at some point a matter of opinions. We do not share but respect the reviewer´s 1 opinion.
The authors significantly improved their manuscript, however, there are still some points which remain unanswered.
We greatly appreciate these words of support
- How the numerical analysis shown in figs 2-5 was performed?
This was detailed in M&M. Quantification of stained areas with Image Pro Plus (Media Cybernetics, Silver Spring, MD, USA) in each image. Further details can be found in our previous publication (ref 19)
How many fields were analyzed and how these fields were chosen.
As mentioned in M&M at least 6 fields per parameter per lesion. Provided that there were 8 animals per group and 1-2 lesions per animal, this equals to 48 to 96 fields per group for each parameter. Fields were chosen randomly. “The two main cross-sections obtained in each implant were considered to be representative of the sample. Three random high-power fields were photographed per cross-section with the use of “LAS V4” software image capture (Leica Mycrosystem, Heerbrugg, Switzerland) system related to a Nikon Eclipse E400 microscope. Therefore, a total of 6 images (1 implant x 2 cross sections x 3 high-power fields) per mouse were acquired for the subsequent determination of each parameter of interest.
- I have a concern regarding statistical analysis and data presentation as a mean and SEM. As I have previously mentioned the nature of the results (proportions) suggests that they should be rather subjected to a non-parametric statistical evaluation. Superficial analysis of SEMs shows that the standard deviation values are high that may argue for a not normal distribution of data. So, authors, please, present median and range for the analyzed groups.
Thank you very much for your observation.
It is true that our results are proportions. There is however the misconception that “proportions” require necessarily a non-parametric analysis for statistical purposes. This misconception arises because commonly these proportions may not adjust to normality, so a non-parametric test would be required. As mentioned in our previous response, we used Sigma Plot for statistical analysis, a software which automatically detects whether data adjust or not to normal distribution as a requisite to authorize or not the test demanded. Anyway, to be absolutely sure we consulted a statistician and performed the normality test (Shapiro-Wilk) in our proportions and all the groups passed the test. Therefore, a parametric analysis, such as ANOVA, can and should be performed despite being proportions.
Anyway we greatly appreciate the reviewer´s comment in this regard as he/she was right that our S.E.M seemed to high. Indeed after careful revision of the data we realized that we had been representing SD instead of SEM. Therefore rather than modifying the graphs to contain the S.E.M we have just left the graphs untouched and modified the text in the manuscript to reflect that SD rather than SEM is represented.
There is also one more thing that requires explanation. The number of coauthors in the present version of the manuscript, which is not considerably different from the former one, is reduced. Does it mean that the authors deleted from the present version (3 names) of the manuscript did not contribute enough? If so, why they were included in the former version?
- Besides of mistakes, as reviewer 1 will understand information on how authorship is handled in the group has to remain confidential and restricted to the research members of the group. Thank you in advance for your comprehension.
Reviewer 2 Report
Main comment
Did the authors consider (1) long-term analyzes and (2) a pain / discomfort investigation? Please discuss it.
Minor comments
line 40: Results warrant > “Our preliminary study opens the way to” is more factual.
line 167: Please clarify how the signal has been normalized (see with line 212) and provide software references.
lane 178: MAB > monoclonal antibody.
line 180: incorrect sentence.
line 197: Please clarify the quantification method: what is the denominator? (The area analyzed? The tissue area? The gland area? Exclusion of the lumen?)
line 202: Did the data meet all the criteria for running parametric analyzes?
line 238: This line is speculative and should be part of the discussion. To introduce the concept, please specify that this is a working hypothesis.
line 279: "In a few pics"> "In several images (n,%),". Same for line 280: "most images (n,%)". This information can be useful in indicating whether the event is rare and found only after extensive work.
Figure 1D: I recommend colorizing the lines and associated symbols. Please double check the symbols (in its current form, this suggests that the controls are statistically very significantly different from themselves).
Figure 2A: The scale bar is not readable. Please improve English in the associated caption.
Figure 3A-C: Scale bar is not readable.
Figure 4D: Here, apoptosis is quantified by the positive area in%. Please specify whether this is the area labeled DAPI or the analyzed field. Please note the variability of the light area in these representative images. The count of apoptotic cells is generally more precise and related to the total number of cells.
Author Response
Main comment
Did the authors consider (1) long-term analyzes and (2) a pain / discomfort investigation? Please discuss it.
AR: Yes of course. We are actually characterizing pain at short and long term. Indeed, we are part of a European consortium devoted to developing pain analysis methods in mouse models of endometriosis. Adequate characterization of pain takes long periods of time and several different evokes and non-evoked pain and thus go beyond the scope of this manuscript paper. Better said we will require a whole independent manuscript to characterize pain endometriosis mouse models to pain in response to D2-ag. Manuscript arising are expected to be jointly published with members of the consortium and would anyway require their previous consent.
Minor comments
line 40: Results warrant > “Our preliminary study opens the way to” is more factual.
Thanks for the suggestion, it is changed in line 42
line 167: Please clarify how the signal has been normalized (see with line 212) and provide software references.
The signal (average radiant efficiency) at each time point was normalized against the time point in which signal was maximal (usually, but not always day 5) in each animal. The software used (living image 4.7.3) is indicated in section “Obtention of raw fluorescent signal” paragraph line 165.
lane 178: MAB > monoclonal antibody.
Changed in line 188
line 180: incorrect sentence.
Modified, see line 190
line 197: Please clarify the quantification method: what is the denominator? (The area analyzed? The tissue area? The gland area? Exclusion of the lumen?)
This is added in line 212
line 202: Did the data meet all the criteria for running parametric analyzes?
Yes. As detailed to reviewer 1. All sets of data groups passed the normality test (Shapiro-Wilk). Errors were represented as SD but we wrongly labeled SEM. This has been addressed accordingly.
line 238: This line is speculative and should be part of the discussion. To introduce the concept, please specify that this is a working hypothesis.
- The statement includes an hypothesis but also a reasoning….(i.e maturation is different to destruction) anyway, line has been moved to the discussion as suggested.
line 279: "In a few pics"> "In several images (n,%),". Same for line 280: "most images (n,%)". This information can be useful in indicating whether the event is rare and found only after extensive work.
- Done as suggested
Figure 1D: I recommend colorizing the lines and associated symbols. Please double check the symbols (in its current form, this suggests that the controls are statistically very significantly different from themselves).
Figure 1 lines were modified as suggested. Symbols related to modifications against basal values were deleted as those measurements had no sense. We apologize for the missunderstanding
Figure 2A: The scale bar is not readable. Please improve English in the associated caption.
Scale changed and English revised.
Figure 3A-C: Scale bar is not readable.
The scale bar was increased. See figure 2-5.
Figure 4D: Here, apoptosis is quantified by the positive area in%. Please specify whether this is the area labeled DAPI or the analyzed field. Please note the variability of the light area in these representative images. The count of apoptotic cells is generally more precise and related to the total number of cells.
- Yes, this represents the area labeled by DAPI. Perhaps these are not the most representative images we agree. We also agree that the count of apoptotic or Ki-67 stained cells is more representative than the area. We had a similar discussion with the reviewers when publishing our previous report.(ref 19) and agreed to state the paragraph here I am copypasting “ , according to previously described methods (Caulet et al. 1992, Moody et al. 2004). We initially aimed to evaluate the proliferation index by quantifying the number Ki67 (proliferation marker)-positive cells per total cell number in each area of interest. However, because of the diffuse (brown) staining, the software failed to automatically segment the number of brown-stained nuclei in the quantification of this parameter. Thus, we decided to evaluate proliferation by also quantifying the area stained by Ki67 per area of interest. Despite not being as accurate as counting individual proliferating cells, quantifying the Ki67-stained area is also a trustable and accepted methodology for the estimation of proliferation indexes, which has been previously used by other investigators (Caulet et al. 1992, Belur et al. 2011)
So at the end we have the macros optimized for counting areas, as long as the areas are adequately quantified we consider measurements are reliable and valid.
Round 2
Reviewer 1 Report
The authors respondjed to all querries and I find their explanations satisfactory. The manuscribt has been significantly improved and is suitable for publication in tys present form.
This manuscript is a resubmission of an earlier submission. The following is a list of the peer review reports and author responses from that submission.
Round 1
Reviewer 1 Report
The paper of Santos-Llamas et al. describes that cabergoline, a dopamine receptor type 2 agonist exerts an antiangiogenic effect and retards growth of experimental human endometrial implants in nude mice. This study is not original as it duplicates the results that has previously been reported by other investigators (Novella-Maestre et al., Hum. Reprod. 24(5):1025-35, 2009; Novella-Maestre et al., Biol Reprod. 83(5):866-73, 2010; Novella-Maestre et al., Fertil Steril. 98(5):1209-17, 2012; Delgado-Rosas et al. Reproduction 142(5):745-55, 2011). In this respect the study adds nothing new to our knowledge on the possible clinical application of dopamine agonists in the treatment of endometriosis.
Generally, the methodology of the study is correct, however I have some specific queries:
- 4% neutral buffered formalin and PFA 4%; isn’t it the same?
- The methods of quantification of the fluorescent signal and quantification of immunofluorescent/immunohistochemical images need more detailed description; e.g. it is not clear whether the authors evaluated an area or the numbers of positive cells.
- The nature of the results (proportions) suggests that they should be rather subjected to a non-parametric statistical evaluation.
Reviewer 2 Report
The manuscript is generally well prepared. Its rational to reassess the effect of dopamine receptor 2 agonists in endometriosis was explained and its methodology is presented adequately.
I suggest accept with minor revision.
Title
Change “D2-AG” to “Dopamine receptor 2 agonist”
Abstract
Line 26 – I am not sure if “non-toxic” is the most suitable word, particularly in the first sentence of the abstract. I suggesting to omit it and change the sentence to “Dopamine receptor 2 agonists (D2-ags) are proposed to reduce endometriotic lesion size by targeting aberrant angiogenesis.”
Line 38 – Change “comparably” to “comparable”
Introduction
Line 55 – Suggest changing “unfortunately” to “However”. If the use of VEGF/VEGFR2 blockers in endometriosis has been vetoed due to the unacceptable toxicity, this should not be “unfortunate” to anyone.
Line 62 – Dopamine receptor 2 agonists (Dr2-ag) – it has been abbreviated as “D2-ag” in the title, the abstract and some parts of the main text. Please make them uniform throughout the manuscript.
Line 67 – Consider changing “our group dared to explore” to “our group explored”
Line 81 – Remove “And” from the beginning of a sentence.
Line 82 – Remove “we reasoned,”
Experimental Section
Line 141 – “control group with 100 μl saline vehicle” What was the route of administration and its frequency?
Line 152 – Delete “)” after raw fluorescent signal
Results
Line 216 – Delete “Anyway,”
Line 219 – What is “said lesion”?
Line 230 – “These results suggest that optimal doses of both D2-ag and anti-VEGF are equally effective in exerting a significant decrease in lesion size.” Only the results of the experiment should be presented in the results section, and not what they suggest.
Line 260 – “These results suggest that very likely both D2-ag and anti-VEGF do significantly interfere with vessel formation in lesions by inducing destruction and/or maturation of immature vessels.” Again, only the results of the experiment should be presented in the results section, and not what they suggest.
Line 303 – “These results suggest that both, anti VEGF and D2-ag treatment induce a similar decrease in the amount of innervating fibers.” Again, only the results of the experiment should be presented in the results section, and not what they suggest.
Line 315 – “serious side effects (i.e thromboembolism, gastrointestinal perforation)” should be changed to “serious side effects such as thromboembolism or gastrointestinal perforation”.
Discussion
Line 316 – “In order overcome” should be “In order to overcome”
Line 317 – “Drd2-ag” This is the third abbreviation for dopamine receptor 2 agonist found in this manuscript. Please correct them.
Line 317 – “the use of Drd2-ag as non-toxic potential selective inhibitors” This suggest that anti-VEGF is “toxic”. Anti-VEGF should not be described as toxic and cabergoline has a list of side effects. Consider changing wording to something different such as less toxic, more benign, or preferable side effect profile.
Line 324 – Remove “do” from “D2-ag do effect”.
Line 331 – Remove “do” from “decreases in staining intensity observed do reflect”
Line 332 – “It is of note that interestingly.” Remove “it is of note that” and simply state “Interestingly the reduction in size…”
Line 348 – Remove “herein”
Line 353 – “On the basis of our observations it is thus likely to conclude that D2-ag is able to exert antiangiogenic actions on ectopic endometrium.” Consider simplifying to “On the basis of our observations it is likely that D2-ag exerts antiangiogenic actions on ectopic endometrium.”
Line 388 – “D2-ag might still be ideal for targeting i) early red lesions exhibiting a high angiogenic activity ii), new lesion formation after surgical removal and iii) lesions with many immature microvessels.” Even if this is true, patient selection (diagnosing such particular lesions over other types of endometriosis) would be a very difficult task.
If the administration of D2-ag needs to be aligned with the window of time in which ectopic endometrial vessels are sensitive to pro-antiangiogenic stimuli, as it did not reduce the overall vascularisation and lesion size when administered 3 weeks after human endometrium were ectopically implanted but it reduced them when administered 5 days after the implantation, it seems to me that its usage in humans would be extremely limited, difficult and unlikely. I would like to see more discussion from the authors in the manuscript.
Line 390 – “Of special relevance in this latter kind are endometrioma lesions in which up to 87% of vessels seem to be of the immature category”. Endometriomas are easily diagnosed with ultrasound and they can adequately be treated surgically. New medical treatment should be directed at the other two phenotypes of endometriosis that are superficial peritoneal endometriosis and deep infiltrating endometriosis.
